# Sonoluminescence Spectra in the First Tens of Seconds of Sonolysis of [BEPip][NTf_2_], at 20 kHz under Ar

**DOI:** 10.3390/molecules27186050

**Published:** 2022-09-16

**Authors:** Rachel Pflieger, Manuel Lejeune, Micheline Draye

**Affiliations:** 1ICSM, Univ Montpellier, CEA, CNRS, ENSCM, F-30207 Bagnols-sur-Cèze, France; 2EDYTEM, University of Savoie Mont Blanc, F-73000 Chambéry, France

**Keywords:** sonolysis, degradation, ionic liquid, viscosity, sonoluminescence

## Abstract

Following recent works on the sonochemical degradation of butyl ethyl piperidinium bis-(trifluoromethylsulfonyl)imide ([BEPip][NTf_2_]), monitoring of sonoluminescence (SL) spectra in the first tens of seconds of sonolysis was needed to better characterize the formed plasma and to question the correlation of the SL spectra with the viscosity. A very dry [BEPip][NTf_2_] ionic liquid (IL) and a water-saturated liquid are studied in this paper. In both cases, IL degradation is observed as soon as SL emission appears. It is confirmed that the initial evolution of the SL intensity is closely linked to the liquid viscosity that impacts the number of bubbles; however, other parameters can also play a role, such as the presence of water. The water-saturated IL shows more intense SL and faster degradation. In addition to the expected bands, new emission bands are detected and attributed to the S_2_ B-X emission, which is favored in the water-saturated ionic liquid.

## 1. Introduction

Ionic liquids (ILs) are molten salts that present melting points less than 100 °C at atmospheric pressure [1]. Their remarkable properties of quasi non-flammability, non-volatility, good thermal [2] and electrochemical [3] stabilities, and recyclability have attracted much attention these last few years, and identify them as a new class of solvents [4,5]. Although they present very low vapor pressure, ILs can be used as solvents for reactions that require the use of ultrasound. Especially, the use of ultrasound in an IL allowed asymmetric catalysis that involved a porphyrin catalyst [6]. A previous study [7] confirmed that cavitation occurs in pyrrolidinium- and piperidinium-based bis-(trifluoromethylsulfonyl)imide ILs. However, the typical signs of slight degradation of ILs by pyrolysis, when used under ultrasonic irradiation, were also observed, precluding the recycling of the catalytic system. A complete study of the degradation products allowed their structural identification, as well as the identification of the mechanism for their formation. The production of sulfites, sulfates, thiocyanates, carbon disulfur, thiophene, and sulfur dioxide from the IL’s anion was reported, as well as the compounds that arise from the pyrolysis of the cation of the IL and subsequent chemical reactions. It was then shown that sonochemical degradation of the IL decreases by a factor two when the IL is saturated with water and is twenty times less important in a water/IL biphasic system. This was justified by the cavitation that preferentially occurs in water because of its lower viscosity and higher vapor pressure. 

Degradation of IL during sonication was also documented by sonoluminescence (SL) spectrometry with a 20 kHz ultrasound under Ar gas. Oxley et al. [8] and Flannigan et al. [9] reported the molecular emission from excited states of C_2_ and CH in the SL spectra of butyl-methylimidazolium chloride and increasing light absorption in the UV. Considering the negligible vapor pressure of ILs that makes their evaporation into the bubble core almost zero, they attributed these emissions to the reactions that occur in a heated shell around the bubbles or inside the bubbles after microdroplet injection.

Similarly, Chatel et al. [7] showed that SL spectra of [C_8_mpyrr][NTf_2_] and [C_8_epip][NTf_2_] ILs (Figure 1) presented molecular emissions from excited states of CN and C_2_. They observed for [C_8_mpyrr][NTf_2_] that increasing the sonication time led to a decrease in the intensities of the SL continuum and lines. This decrease was attributed by Pflieger et al. [10] to the progressive poisoning of the plasma inside the bubbles by the sonolysis degradation products. This latter work presented the time evolution with a one-minute resolution of SL spectra of a very dry [BEPip][NTf_2_] ionic liquid (Figure 1) under Ar and underlined that degradation of the IL had already started in the first minute of sonication. It resulted in rovibronic temperatures for C_2_ and CN excited species (vibrational temperatures of 5800 ± 500 K and 6000 ± 500 K for C_2_ and CN, respectively, and rotational temperatures of 4000 ± 500 K), showing that they were surprisingly constant during the sonolysis, despite the undergoing poisoning of the plasma. It also suggested a correlation between the intensities of the SL continuum and molecular emissions and the IL viscosity, controlled by the IL temperature. Indeed, the latter strongly increased at the beginning of sonication, reaching its steady-state value after two minutes, and leading to a fast decrease in the viscosity, favoring bubble nucleation [11] and the formation of a higher number of sonoluminescing bubbles. Hence, there was a significant increase in SL intensities observed in the first two min of sonication.

To further understand the IL degradation under ultrasound and in the characterization of the formed plasma, the present work aimed to acquire SL spectra in the very early stages of sonication, when the degradation of the IL is minimal, and question its correlation with the viscosity. To increase the time resolution, the SL intensity needed to be increased, and the initial temperature was raised to 25 °C to lower the viscosity and favor cavitation. In addition, since it was previously [7] shown that the presence of water protected the IL from degradation and while thorough drying prior to sonication leads to intense SL, both a very dry [BEPip][NTf_2_] IL and a water-saturated liquid were studied here with regard to their SL time evolution in the very first moments of sonication.

## 2. Materials and Methods

The IL [BEPip][NTf_2_] was prepared and characterized as described previously [10]. It was then used in its water saturated form or dried under a vacuum and stirred at 80 °C for one and a half days to keep the water content as low as possible. Viscosity was measured using a Lovis 2000 M Anton Paar microviscosimeter.

A volume of 30 mL IL was placed in the home-made stainless-steel reactor [7] equipped with a 20-kHz horn (1 cm^2^, 750 W Sonics generator), which was set to work at 35% of its maximum amplitude (leading to an estimated absorbed acoustic power P_ac_ of 15 W) under an Ar (99.999% purity, Air Liquide) flow of 14 mL/min. The Ar flow started half an hour before sonication and continued during it. The temperature of the reactor and the contained IL was controlled by setting the Huber Unistat Tango thermo-cryostat at 25 °C and monitored during sonication. Although lower temperatures are usually preferred to enhance sonoluminescence intensity, a higher target temperature was chosen in order to decrease the viscosity of the IL, and thus favor cavitation. Thus, the IL initial temperature was 25 °C and then increased during the sonolysis due to cavitation.

Sonoluminescence spectra were collected through a flat quartz window using parabolic Al-coated mirrors and recorded with a SP 2356i Roper Scientific spectrometer, coupled with a CCD camera with UV coating (SPEC10-100BR or PyLoN:400BR_eXcelon, Roper Scientific) cooled to −120 °C by liquid nitrogen. As the emitted light was very dim, the first attempts focused on amplifying it and an emission spectrometer coupled to an intensified camera (I-CCD) was used. Unfortunately, this attempt was not successful because the noise (the scatter in the signal) was amplified to the same extent as the signal itself. Thus, the best technical solution remained the use of a liquid nitrogen cooled CCD and the lower the camera temperature (−120 °C), the lower the noise in the measurement of the SL light intensity. A low-resolution grating (150blz500; spectral resolution 2.8 nm) was used to measure the whole spectrum (from 320 nm up to 850 nm) at once. Spectral calibration was performed using a Hg(Ar) pen-ray lamp (model LSP035, LOT-Oriel) and correction for background noise and for the quantum efficiencies of grating and CCD was carried out. Measurement of the SL spectra started at the same time as sonication. Spectra were measured every 7 s with an acquisition time of 5 s or every 17 s with an acquisition time of 15 s.

All molecular emissions were attributed based on the reference book *The Identification of Molecular Spectra* [12].

## 3. Results and Discussion

### 3.1. Follow-Up of Temperature and Estimation of Viscosity

The evolution of the IL viscosity in the first moments of sonolysis is needed to further explore the link between SL intensity and viscosity of the IL. In the first approximation, the temperature dependence of the viscosity of [BEPip][NTf_2_] can be estimated using the following formula derived for ethyl-methylimidazoliumfluoride by Liu et al. [13]: η = 2.07 × 10^−7^ exp (−Ea × 10^4^/(R.T))(1)
where Ea is the corresponding activation energy, R is the rare gas constant and T is the temperature. 

The viscosities of the dry and water-saturated ILs before sonication were measured at 25 °C and were 216 and 188 mPa.s, respectively. Inserting these values into Equation (1) gives activation energies of −51.5 kJ/mol for the dry IL and −51.1 kJ/mol for the water-saturated one.

The temperature evolution measured during the sonolysis of dry [BEPip][NTf_2_] is plotted in Figure 2, together with the viscosities calculated using Equation (1) for both ILs. The temperature evolution of the hydrated IL is assumed to be very similar due to very close C_p_ values of both ILs. The temperature shows a fast increase in the first tens of seconds, reaching 41.5 °C after 40 s, then plateaus around 45 °C. Consequently, the viscosity significantly decreases in the first 40 s of sonolysis and reaches a steady-state value after 60 s, at about one fourth of its initial value. It should be noted that the viscosity of the water-saturated IL is lower than the one of the dry IL throughout the sonication process, but the difference becomes smaller after 30 s.

### 3.2. Time Evolution of the SL Spectra of the Water-Saturated IL with 15 s Acquisition Time

Appendix A in the Supporting Information presents the general evolution of the SL spectra of a water-saturated [BEPip][NTf_2_] sonicated at 20 kHz under Ar flow. Only one spectrum for every ten spectra is presented for clarity. As previously reported [10], progressive coloring of the IL occurs, leading to strong absorption of the SL light in the UV and visible domains. Molecular emissions are present on top of the SL continuum from the beginning of sonication. 

To exemplify the evolution of the SL continuum and of the molecular emissions, SL spectra corresponding to the sonication times of 15 and 32 s (resp. 49, 66 and 83 s, resp. 100, 117, 134 and 151 s) are depicted in Figure 3a (resp. b, resp. c). In the first two spectra, the SL continuum is relatively low and emissions from CN violet and C_2_ Swan bands are clearly visible, which were also previously observed in the sonolysis of [BEPip][NTf_2_] and that arise from the degradation of the cation. Small emissions can also be noticed around 673 nm that may be attributed to the second-order light emission of NH A-X. After 32 s of sonication, the SL intensity strongly increases and reaches a maximum. New emission bands appear that are particularly clear in the 4th and 5th spectra. They are indicated by red lines in Figure 3b and correspond to S_2_ B-X emission. This emission was not previously reported in the sonolysis of [BEPip][NTf_2_], probably because it is not very intense in the very first spectra and is soon hidden by absorption as degradation progresses. It is to be noted that the SO B-X transition (SO’s most intense transition), which would be expected from degradation of the anion, cannot be observed in the present experimental conditions, since its main lines are below 330 nm, and the used grating is not adapted to this wavelength range. The same limitation holds for CF and CF_2_ transitions (<322 nm). As for the SO_2_ emission, it cannot be ruled out but would be superimposed with the very intense CN transition (around 387 nm).

Diatomic sulfur can be obtained by thermal decomposition of hydrogen sulfide at temperatures higher than 1000 °C [14]. From an algorithm they developed, Kaloidas and Papayannakos [15] showed that the mixtures produced from H_2_S decomposition are mainly composed of diatomic sulfur molecules, which constitute more than 99.8% of the sulfur-bearing molecules in the temperature range of 973–1123 K and pressure range of l–4 atm. These temperatures and pressures are in good agreement with those observed in the immediate environment of the bubble at the moment of its implosion. 

The formation of diatomic sulfur in its excited state may also arise from reactions in the gas phase, by recombination of S atoms via a three-body reaction, which is as follows: S + S + M → S_2_ + M(2)

This reaction has been described for the third body (M) being Ar [16,17] and H_2_S [18]. As for the formation of elemental sulfur, it has been reported in volcanic gases containing SO_2_ and H_2_S [19,20,21] (reaction (3)), and in pulse discharges in H_2_S gas [18] (reaction (4)), in H_2_S-H_2_-Ar gas mixtures [17] (reactions (5) and (6)) and in SO_2_-H_2_-Ar gas mixtures [22] (reactions (7) and (8)).
SO_2_ + 2 H_2_S ↔ 3 S + 2 H_2_O(3)
H_2_S → 2 H + S(4)
H + H_2_S ↔ H_2_ + HS(5)
H + HS ↔ H_2_ + S (6)
2 H + SO_2_ → H_2_O + SO(7)
2 H + SO → H_2_O + S(8)

The formations of both SO_2_ and H_2_S gases have been reported in the sonochemical degradation of similar ILs ([C_8_mpyrr][NTf_2_], [C_8_epip][NTf_2_] and [BEPip][NTf_2_]) [7]. As discussed above, S_2_ can a priori be formed from these gases both in the gas phase and at the bubble–liquid interface. 

Interestingly, the formation of S_2_ during the sonolysis of [NTf_2_]-based ILs was not observed previously, neither in SL emission spectra nor in the macroscopic quantification of the degradation products. This absence (or presence in a smaller amount) may be attributed to the presence of water in the water-saturated ionic liquid, while mostly dry ionic liquids were previously studied (water content < 50 ppm [7] or even < 10 ppm [10]). Indeed, the following reaction of SO_2_ with water also leads to the formation of S atoms [21]:3 SO_2_ + 3 H_2_O → 2 H_2_SO_4_ + H_2_O + S(9)

Figure 4 presents the evolution of the intensities of CN, C_2_, NH and S_2_ molecular emissions and of the SL continuum taken at 491 nm (for all intensities, after correction for the absorption); when several bands are visible for a particular excited species, the most intense one was considered. For all these intensities, a strong sudden increase was observed after 30 s of sonication, followed by a quasi-steady state or slow decrease, in agreement with the previously reported data [10]. This trend can be explained by the evolution of the viscosity (Figure 5). Indeed, the latter strongly decreases in the first 30 s, reaching one third of its initial value. This lower viscosity favors nucleation and the formation of a higher number of cavitation bubbles, hence the increase in SL intensity. Its evolution is then much slower, which reflects the similarly less important evolution of the SL intensity. As for the observed decrease in the SL intensity while the viscosity is approximately constant, it is attributed to the poisoning of the plasma inside the bubbles by IL droplets or by volatile degradation products.

In general, the observed phenomena display (much) faster kinetics than in our previous study [10], where the continuum intensity was observed to decrease only after 5 min of sonication. This different behavior is due to the difference in the initial temperature. Indeed, the initial temperature is higher in the present study compared to the previous one (25 °C vs. 1 °C), resulting in a 6-times lower initial IL viscosity according to Equation (1). The subsequent higher number of cavitation bubbles in the first seconds of sonication leads to the faster degradation of the IL; a strong decrease in the SL intensity was previously observed after 5 min (with the thermocryostat set temperature being 1 °C), while with the higher thermocryostat temperature (25 °C), the SL intensity stops increasing after 1 min.

### 3.3. Time Evolution of the SL Spectra of a Very Dry IL with 5 s Acquisition Time

A very dry IL was then used to increase the SL intensity to its maximum so as to decrease the acquisition time as much as possible. To monitor in more detail the evolution of the SL continuum and the molecular emissions within the first minute of sonication, the acquisition time was reduced to a minimum (5 s) and no averaging was performed. Evidently, this leads to a larger scatter in the signal. Figure 6 presents the evolution of the corresponding SL spectra in the first moments of sonication. For clarity, to avoid superimposition, the spectra in Figure 6 were plotted in the stack mode, i.e., shifting them on the *y*-axis.

SL light is hardly distinguishable in the first spectrum, contrary to the water case, where cavitation develops after a few ms [23]. This difference is traced back to the high viscosity of [BEPip][NTf_2_] that leads to the (very) slow development of the cavitation pattern, as reported by Tzanakis et al. for glycerine (ν = 934 mPa s at 25 °C [24]) at 20 kHz [25].

Sonoluminescence is, however, clearly detected in the second spectrum (corresponding to a sonication time of 7 to 12 s). Interestingly, this second spectrum does not only feature the SL continuum, but also a molecular emission around 390 nm, corresponding to the CN emission (similar to what was observed with a 15-s acquisition time), indicating that the degradation of the IL has already taken place. In addition, similar to the previously published work and to spectra measured with a 15-s acquisition time, no Ar emission was observed, neither in the first spectrum, where SL (if any) was hardly detected, indicating that cavitation may have not started yet, nor in the second spectrum that bears signs of IL degradation, whereby degradation products can quench excited argon atoms. These observations all indicate that degradation occurs from the very beginning of cavitation.

This early sign of degradation may seem surprising, considering that the high initial viscosity of the IL is expected to dampen non-linear oscillations [26], extend the region of shape stability [27], reduce translational motion [26,28], limit jetting and enhance stability against splitting [25,29,30], thus restricting droplet injection, which is one possible origin of the degradation of the non-volatile IL. This mechanism cannot, however, be ruled out: on the one hand, viscosity attenuates bubble growth; on the other hand, it results in a broader zone of shape stability [27], which allows bubbles to be stable in their shape (and SL emitting) at higher acoustic pressures, which results in larger bubble radii. Thus, larger sonoluminescing bubbles can be present in viscous media compared to water, as was reported in sulfuric acid at 23 kHz [31]. This larger size resulted in high primary Bjerknes forces, leading to fast translation of the bubbles and subsequent jetting, thus resulting in the introduction of liquid into the bubble core. Considering that the viscosity of [BEPip][NTf_2_] is almost one order of magnitude larger than that of sulfuric acid (23.8 mPa.s at 25 °C), similar jetting and droplet injection processes may be expected. In addition, secondary Bjerknes forces are impacted by the viscosity, and larger bubbles result in stronger secondary Bjerknes forces and in stronger interactions between bubbles. It should be noted that a high liquid viscosity leads to the formation of peculiar bubble structures [32,33] that may also impact jetting. Ban and Choi [34] observed the formation of bubble clusters in Kr-saturated 80% glycerol (ν ≈ 100 mPa s) sonicated at 50 kHz that were similar to the clusters reported to emit Na-atom emissions and exhibited droplet injection in Xe-saturated NaCl-ethylene glycol (ν ≈ 15.5 mPa s) at 28 kHz [35]. 

The second location of IL degradation can be found at the bubble–liquid interface. Indeed, Burrell et al. demonstrated the limited thermal stability of imidazolium- and pyrrolidinium-based ionic liquids [36]. In their work, degradation products were shown to be already formed by heating butylmethylpyrrolidinium- and butylmethylimidazolium bis-(trifluoromethylsulfonyl)imide ILs at temperatures above 150 °C. The temperature at the bubble–IL interface at the moment of a bubble implosion is expected to be (much) higher than 150 °C (higher than in less viscous solvents due to heating by viscous dissipation [37]), and can, thus, explain the early degradation of the IL observed in the present work.

Figure 7 presents the evolution of the SL continuum intensity for the dry IL. In the first 15 s, the trend is as expected from the liquid viscosity; the SL intensity strongly increases following the concurrent fast increase in temperature that strongly decreases the viscosity and leads to an increase in the number of SL bubbles. A first maximum in the SL continuum intensity is then observed (in the third spectrum), followed by a first decrease until 27 s, while the viscosity continues decreasing, which should favor cavitation. The SL intensity then increases again up to its second maximum (at 55 s), although the viscosity has almost reached its steady-state value, before decreasing again. This time evolution of the SL continuum intensity does not show the monotonous trend that was previously [10] reported for a 1-min time resolution, during which all the measured intensities strongly increased during the first two minutes of sonication. Similarly, the spectra acquired here with a 17-s time resolution on a water-saturated IL showed an increase in the continuum intensity, followed by a slow decrease. It is interesting to compare the [BEPip][NTf_2_] case to that of glycerol aqueous solutions because of their similar viscosities. Ban and Choi [34] observed that the 50 kHz SL intensity of glycerol solutions saturated with Kr increased from 0 to 80% glycerol, then decreased to pure glycerol, which they attributed to the too strong damping of the bubble oscillations at the high viscosity of pure glycerol (934 mPa.s). It is also at 80% glycerol that bubble clusters were reported, clusters that were correlated to the line emission in SL. No other intermediate concentrations between 60% glycerol (20.4 mPa.s) and pure glycerol were investigated that would allow more accurate comparison of the IL with glycerol solutions. Thus, it seems that in the interval covered by the IL viscosity during sonolysis (around 100 mPa.s), several changes occur in the bubble population and dynamics, and their interplay, coupled to IL degradation and plasma poisoning, may explain the presence of the two maxima in Figure 7. 

Figure 8 presents the SL spectra measured at 41 s and 48 s, whereby the latter corresponds to the maximum intensity of SL. The same molecular emissions as those obtained with a 17-s time resolution can be observed, although they are less intense, and include C_2_ Swan bands, CN violet, CH A-X, the second order emission of NH A-X and S_2_ B-X. The time evolution of these molecular emissions (Figure 9) follows approximately the same trend as that of the SL continuum with two local maxima, at around 14 s and 55 s. This correlation can be explained by an increase in the number of active bubbles, at the interface of which the IL can be thermally degraded, and possibly leading to an increase in the number of bubbles that present droplet injection. Evidently, the proportionality is not exact (Appendix A); the jetting bubbles will contribute more to molecular emissions, while spherically collapsing bubbles emit a stronger continuum [31].

### 3.4. Comparison of Very Dry and Water-Saturated ILs

This non-monotonous evolution of the SL intensity for the dry IL is not the only difference with the water-saturated liquid. Indeed, contrary to what was expected, the SL intensity is less intense than for the water-saturated IL; the continuum intensity at 491 nm peaks around 130 A.U. for the dry IL vs. 4000 A.U. for the water-saturated IL. The different acquisition time (×3) and type of CCD used (×4) only account for a 12 factor difference. Thus, the water-saturated IL is approximately three times more luminous. One may tentatively explain this by the slightly lower viscosity, combined with the higher vapor pressure that favors nucleation. In addition, the surface tension (that also affects nucleation) can be different in both cases: it is known to depend on slight changes in temperature and surface characteristics or viscosity of the liquid, which can disrupt the equilibrium maintained by the interfacial forces [38]. 

This higher SL intensity in the water-saturated IL indicates the presence of a higher number of cavitation bubbles, leading to the faster initial degradation of the IL. This degradation is exemplified by the stronger light absorption for the water-saturated IL and by the higher intensities relative to the SL continuum of CN and C_2_ (Figure 4, Appendix A) in the 40–100 s time interval. It also explains the fact that the maximum value in SL intensity was reached earlier compared to the dry IL. Interestingly, the intensities relative to the SL continuum of CH and NH were lower in the water-saturated IL, and the SL continuum intensity decreased more slowly, which may indicate different degradation pathways and probably different contributions of the degradations at the bubble interfaces and inside the bubbles. Indeed, an enrichment in water is expected at the interface, due to its low affinity for the IL. The presence of water will locally decrease viscosity, which, combined to water, means that higher C_p_ will decrease the interface temperature. Hence, a lower degradation rate at the interface is observed. 

As for the non-monotonous evolution of the SL intensity of the dry IL in the first 40 s of sonication, it may originate in the small number of cavitation bubbles. Indeed, if only very few bubbles are present, almost no averaging is carried out and the fluctuations are measured. Jetting and/or formation of gaseous degradation products lead to plasma poisoning and SL intensity decrease. Due to the concomitant increase in temperature and decrease in viscosity, new bubbles are formed and the SL intensity increases again. These fluctuations that are related to the smallness of the considered bubble group also reflect the large fluctuations observed in the intensities of molecular emissions relative to the SL continuum (Appendix A).

## 4. Conclusions

The measurement of SL spectra of [BEPip][NTf_2_] during the first tens of seconds of its 20-kHz sonolysis under Ar allows interesting comparisons to be made. First, SL is observed only after several seconds of sonication; much more time is needed to install cavitation compared to water, due to the IL’s high viscosity. Second, inceptions of SL emission and IL degradation are concomitant, as indicated by the presence of molecular bands from the first measurable SL spectrum. Third, the comparison of a very dry IL and a water-saturated liquid confirmed that the initial evolution of the SL intensity is closely linked to the liquid viscosity, especially for the dry IL, but that other parameters can also play a role.

The water-saturated IL produces a more intense SL, traced back to the higher number of active bubbles. This higher number of bubbles also leads to faster degradation and more intense CN and C_2_ emissions.

The possible location of the IL degradation is discussed, and both the bubble interface and the bubble interior are shown to be probable contributors. The contributions of the two locations of degradation are different for the dry IL and the water-saturated IL; the presence of a water layer at the bubble interface induces a decrease in temperature, due to the high Cp of water and local decrease in viscosity, thus reducing thermal degradation at the interface.

Finally, due to the higher time resolution, a new product of the degradation of [BEPip][NTf_2_] was detected, namely S_2_, that can be formed from H_2_S and/or SO_2_ either at the bubble interface or in the sonochemical plasma and was formed in higher amounts in the water-saturated IL.

## Figures and Tables

**Figure 1 molecules-27-06050-f001:**
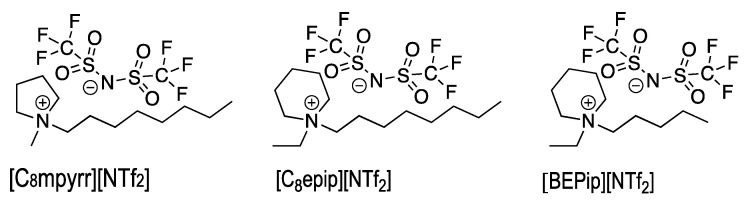
Pyrrolidinium- and piperidinium-based bis-(trifluoromethylsulfonyl)imide ionic liquids.

**Figure 2 molecules-27-06050-f002:**
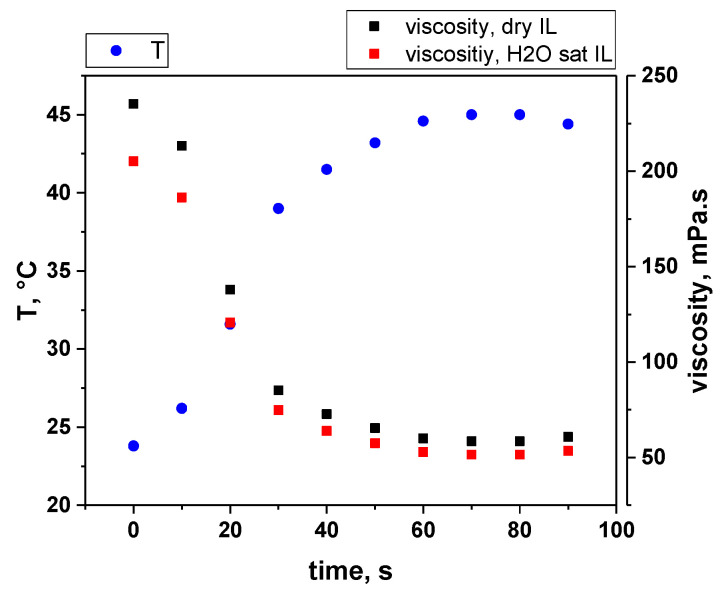
Temperature evolution of the sonicated dry [BEPip][NTf_2_] and corresponding calculated values of the viscosities of dry and water-saturated IL.

**Figure 3 molecules-27-06050-f003:**
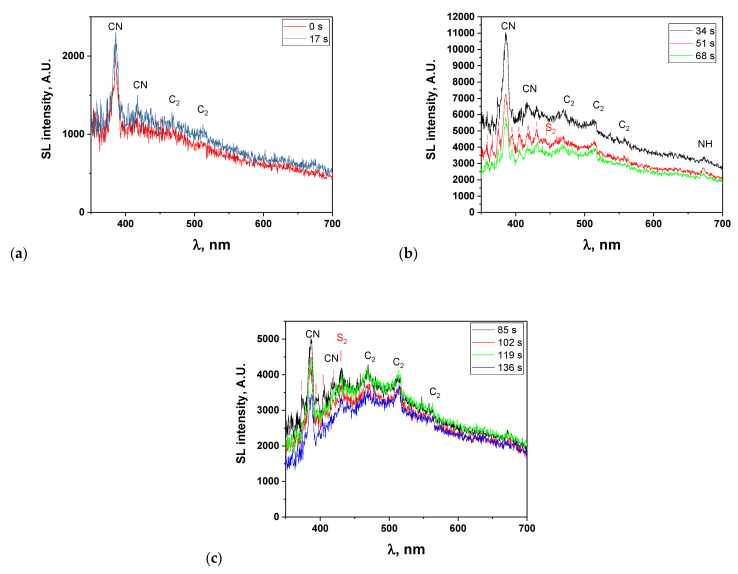
SL spectra obtained in the very first minutes of sonication; indicated times stand for the beginning of the corresponding spectrum (water-saturated [BEPip][NTf_2_] IL, 20 kHz, Ar): (**a**) 0 and 17 s, (**b**) 51 and 68 s, (**c**) 85, 102, 119 and 136 s.

**Figure 4 molecules-27-06050-f004:**
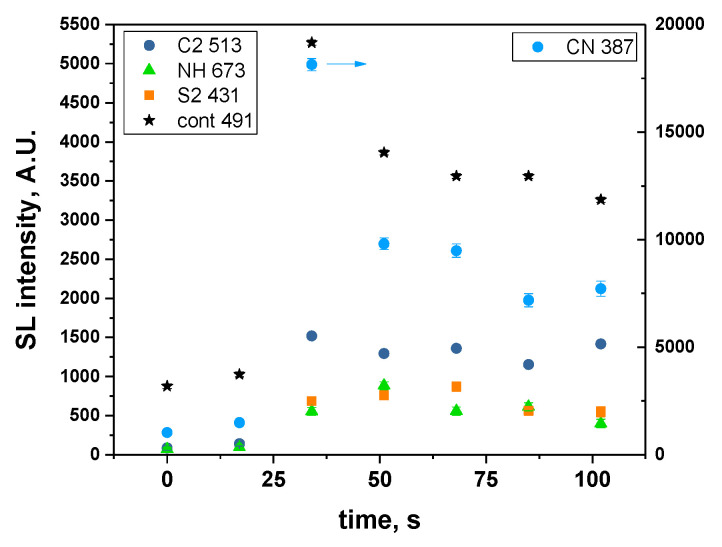
Time-evolution of molecular emissions and of the SL continuum at 491 nm (water-saturated [BEPip][NTf_2_], 20 kHz, Ar), corrected for absorption.

**Figure 5 molecules-27-06050-f005:**
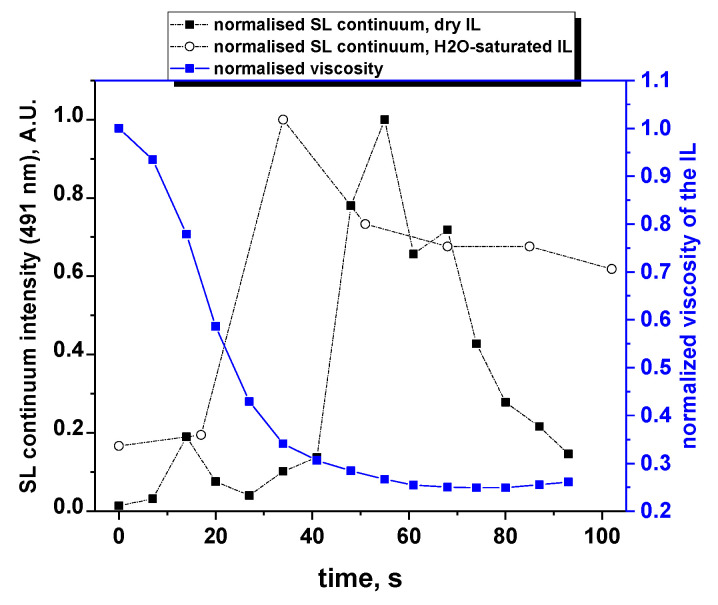
Evolution of the SL continuum intensity and of the estimated viscosity of the IL during the first moments of sonolysis.

**Figure 6 molecules-27-06050-f006:**
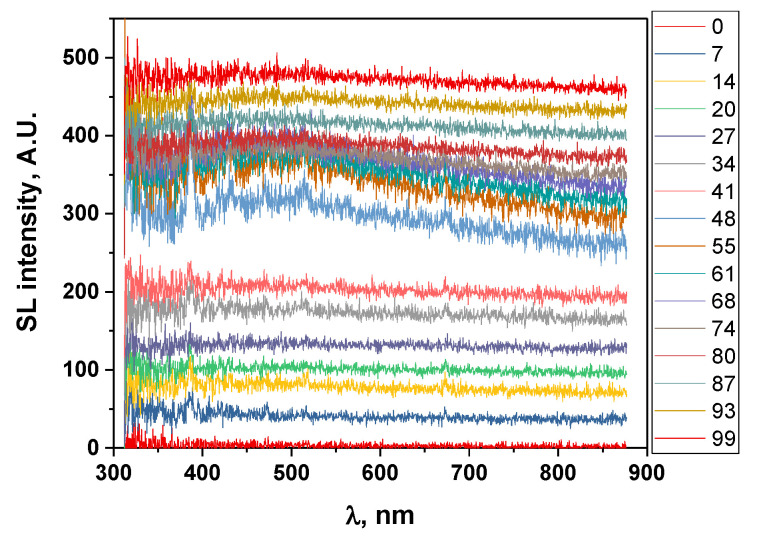
Evolution of the SL spectra of [BEPip][NTf_2_] during the first tens of seconds of sonolysis at 20 kHz under Ar (spectra plotted in a stack mode, with an offset of 30, to avoid being superimposed and to allow some clarity of the graph). Numbers in the caption on the right indicate the starting time (in seconds) of each spectrum (t = 0 corresponds to the onset of sonication).

**Figure 7 molecules-27-06050-f007:**
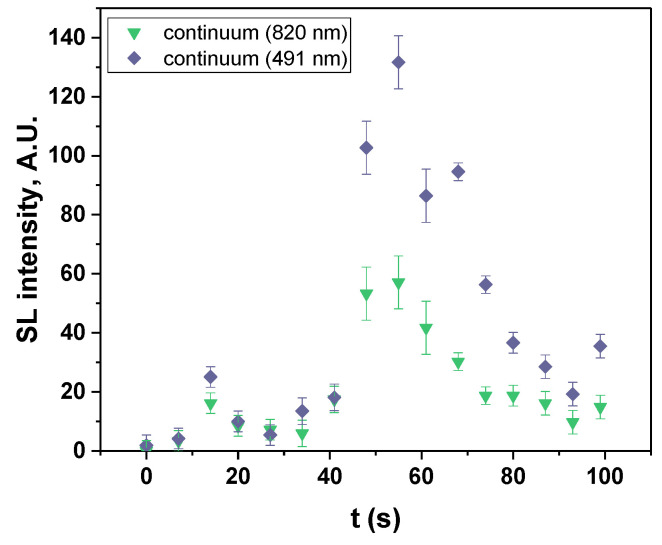
Evolution of the SL continuum intensity (taken at 491 and 820 nm) with time (dry IL).

**Figure 8 molecules-27-06050-f008:**
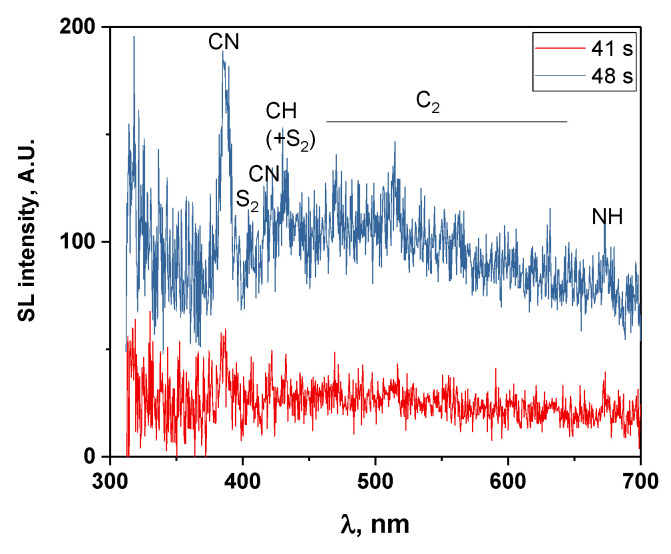
SL spectra of dry IL measured at 41 and 48 s of sonolysis.

**Figure 9 molecules-27-06050-f009:**
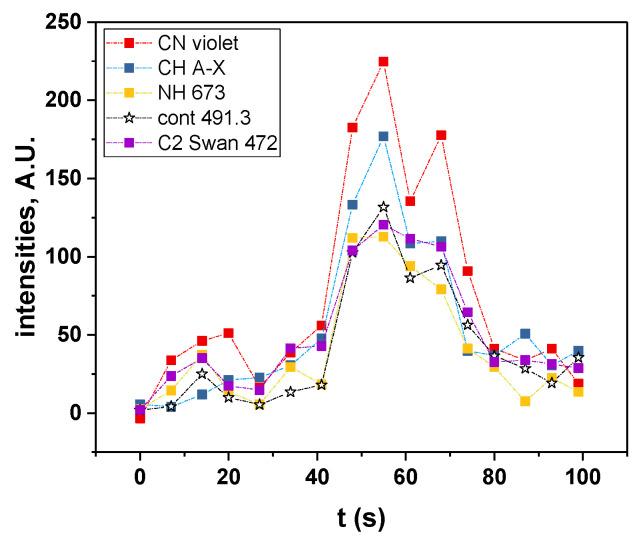
Evolution of the intensities of the main molecular emissions with time (dry IL).

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
