# Peer review of "Sonoluminescence Spectra in the First Tens of Seconds of Sonolysis of [BEPip][NTf2], at 20 kHz under Ar"

_molecules, 2022, doi:10.3390/molecules27186050_

Round 1

Reviewer 1 Report

The work entitled "Sonoluminescence spectra in the first tens of seconds of sonolysis of [BEPip][NTf2], at 20 kHz under Ar". The work presents an interesting description of the degradation of an ionic liquid studied by SL. However, some points should be included to give the work more clarity and that it meets the journal's standards.

a) A general diagram of the degradation route must be included as a function of the temperature for the ionic liquid studied

b) Indicate the half-life times of the fragments proposed for the ionic liquid studied and if this is consistent with the detection limit of the equipment used for the measurement of the spectra.

c) The authors should mention why it was not possible to measure the viscosities during the experiments.

d) There is some other experimental method in the literature that could verify the proposed degradation processes.

Author Response

We thank Reviewer 1 for his/her valuable comments.

  • “A general diagram of the degradation route must be included as a function of the temperature for the ionic liquid studied”

With all due respect we disagree with the referee that a general diagram of the degradation route must be included as a function of the temperature for the ionic liquid studied. Indeed, the determination of the degradation route of the ionic liquid as a function of the temperature is not the subject of the study. In fact, this work is already published (ACS Sustainable Chem. Eng.  2013, 1, 137-143, doi: 10.1021/sc300068d). The present work aims at questioning the correlation of SL spectra with the viscosity, of both a very dry [BEPip][NTf2] and of a water-saturated one at 25°C.

  • “Indicate the half-life times of the fragments proposed for the ionic liquid studied and if this is consistent with the detection limit of the equipment used for the measurement of the spectra.”

With all due respect, we disagree with the referee that the half-life times of the fragments proposed for the ionic liquid studied should be mentioned and if this is consistent with the detection limit of the equipment used for the measurement of the spectra. Indeed, the early degradation of the ionic liquid is characterized by molecular emissions, which are attributed on the basis of the reference work "The Identification of Molecular Spectra" and not by the identification of free radicals. In addition, all identifications are perfectly consistent with the detection limits of the CCD camera cooled with liquid nitrogen.

  • “The authors should mention why it was not possible to measure the viscosities during the experiments.”

We disagree with the referee that it should be mentioned why it was not possible to measure the viscosities during the experiments. Indeed, the equipment used for the described experiments does not allow this kind of simultaneous measurements. It would be the topic of an entire devoted study to develop a set-up allowing to monitor SL spectra and viscosity in parallel at the used time resolution.

  • There is some other experimental method in the literature that could verify the proposed degradation processes.

We agree with the referee that some other experimental methods in the literature could verify degradation processes. This is already published in previous works (ACS Sustainable Chem. Eng.  2013, 1, 137-143, doi: 10.1021/sc300068d) and not the scope of the present study.

We would like to thank again the referee for his/her comments and hope that the revised version is now acceptable for publication in Molecules.

Reviewer 2 Report

Comments on manuscript “Sonoluminescence spectra in the first tens of seconds of sonolysis of [BEPip][NTf2], at 20 kHz under Ar”.

This paper discussed sonoluminescence from RTIL within the first tens of seconds of sonication. The topic is interesting and the derived findings are scientifically sound. However, I have some serious concerns regarding both the content and language quality, which are listed as follows. I recommend reconsideration after major revision.

1.      My general impression after reading the manuscript is that it lacks proper proofreading. The language is not concise and difficult to read. There are numerous issues related to the abbreviation of technical terms, citation, grammar, and sentence structure. Please further polish your paper to the publishable level.  

2.      Section. 2. The experimental setup is very unclear. Please give more details on the apparatus and test procedure.

3.      Data reliability. As is well known, the initial phase of any test is unstable. Transient effects may arise in this stage. However, the whole discussion focuses on this phase. Please prove your data is repeatable and reliable.  

4.      In line 85, the author states that the temperature of the IL is controlled at 25℃. Then how does the temperature increase as shown in Fig.2. Is the temperature controlling unit stopped during sonication? This is just an example of the many similar issues appearing in the manuscript. Please clarify the test condition for each dataset you analyzed.

5.      Line 122, if temperature control is not used in the test, why does the temperature plateau at 45℃?

6.       Line 256-259. The author analyses how high viscosity may lead to degradation of IL by forming large sonoluminescing bubbles which are stable under high acoustic pressures. However, in Fig.5, the viscosity of dry IL at 14 s barely changes from the initial value. If degradation appears at this instant, I would argue similar degradation would occur at 0 s according to the same reasoning. In addition, I doubt at high viscosity, the bubble can be large enough to be able to make translational move.

7.      It seems from Fig.6 that the molecular emission around 390 nm disappears at 99s (the top spectrum line). Please explain this phenomenon.

Author Response

We thank Reviewer 2 for his/her valuable comments.

  1. “My general impression after reading the manuscript is that it lacks proper proofreading. The language is not concise and difficult to read. There are numerous issues related to the abbreaviation of technical terms, citation, grammar, and sentence structure. Please further polish your paper to the publishable level.”

We thank the review for his/her careful reading. The grammatical errors have been carefully addressed in the attached new version of the manuscript. In particular, consistency of the use of abbreviations has been improved. In addition, a native speaker has checked and corrected the remaining mistakes.

  1. “Section 2. The experimental set-up is very unclear. Please give more details on the apparatus and test procedure.”

The experimental set-up is described in details in Ref.6 and its SI. Following the recommendation of the reviewer, we’ve nevertheless added some more details: “The temperature of the reactor and the contained IL was controlled by setting the Huber Unistat Tango thermo-cryostat at 25°C and monitored during sonication” and “Thus, the IL initial temperature is 25°C and then increases during the sonolysis due to cavitation”.

  1. “Data reliability. As is well known, the initial phase of any test is unsuitable. Transient effects may arise in this stage. However, the whole discussion focuses on this phase. Please prove your data is repeatable and reliable.”

This work belongs to research activies in our labs focussing on the degradation of ILs under US. In the past studies, the degradation after hours was reported, together with the various formed degradation products. The evolution of SL spectra was also reported but with a time-resolution of 1 minute. The novelty of the present study is the demonstration of the feasibility of measurement of SL spectra in the very first moments of sonication, which allowed to bring to light that degradation starts to occur from the onset of cavitation. We agree that transient effects can arise in this initial stage, and that’s what we report here as “fluctuations”. However, it is also noticeable that present results agree with previous ones.

  1. “In line 85, the author states that the temperature of the IL is controlled at 25°C. Then how does the temperature increase as shown in Fig. 2? Is the temperature controlling unit stopped during sonication? This is just an example of the many similar issues appearing in the manuscript. Please clarify the test condition for each dataset you analysed.”

Some more details have been added in the experimental section to clarify this point (see question 2). The cryostat temperature is set at 25°C and the cryostat is kept on during sonication. So, initial temperature of the IL is 25°C. As soon as cavitation takes place, the IL is heated by it, because a large part of the acoustic energy is transformed into heat (as is well known in the US community).

  1. “line 122, if temperature control is not used in the test, why does the temperature pleateau at 45°C?”

Temperature plateauing is due to the cooling from the cryostat and the heating from cavitation. A sentence has been added in section 2: “Thus, the IL initial temperature is 25°C and then increases during the sonolysis due to cavitation.”

  1. “line 256-259. The author analyses how high viscosity may lead to degradation of IL by forming large SL bubbles which are stable under high acoustic pressures. However, in Fig. 5, the viscosity of dry IL at 14 s barely changes from the initial value. If degradation appears at this instant, I would argue similar degradation would occur at 0 s according to the same reasoning. In addition, I doubt at high viscosity, the bubble can be large enough to be able to make translational move.”

We believe there is some misunderstanding here. We agree with the reviewer that similar degradation occurs at 0 s indeed, and this is stated in the manuscript at several occurrences for instance in the abstract: “IL degradation is observed as soon as SL emission appears”.

Concerning the translational move, it is something observed by other authors at high viscosity. However, it is only one phenomenon that could explain the jetting of bubbles and subsequent degradation of IL inside cavitation bubbles, in particular due to stronger secondary Bjerkness forces and/or pecular bubble structures formed. This is discussed in 3.3.

  1. “It seems from Fig.6 that the molecular emission around 390 nm disappears at 99 s (the top spectrum line). Please explain this phenomenon.”

The reviewer is right, this emission appears not to be visible any more after a certain time. This is due to light absorption, as indicated in 3.2: “progressive colouring of the IL occurs, leading to strong absorption of the SL light in the UV and visible domains”.

We thank Reviewer 2 for his/her valuable comments.

  1. “My general impression after reading the manuscript is that it lacks proper proofreading. The language is not concise and difficult to read. There are numerous issues related to the abbreaviation of technical terms, citation, grammar, and sentence structure. Please further polish your paper to the publishable level.”

We thank the review for his/her careful reading. The grammatical errors have been carefully addressed in the attached new version of the manuscript. In particular, consistency of the use of abbreviations has been improved. In addition, a native speaker has checked and corrected the remaining mistakes.

  1. “Section 2. The experimental set-up is very unclear. Please give more details on the apparatus and test procedure.”

The experimental set-up is described in details in Ref.6 and its SI. Following the recommendation of the reviewer, we’ve nevertheless added some more details: “The temperature of the reactor and the contained IL was controlled by setting the Huber Unistat Tango thermo-cryostat at 25°C and monitored during sonication” and “Thus, the IL initial temperature is 25°C and then increases during the sonolysis due to cavitation”.

  1. “Data reliability. As is well known, the initial phase of any test is unsuitable. Transient effects may arise in this stage. However, the whole discussion focuses on this phase. Please prove your data is repeatable and reliable.”

This work belongs to research activies in our labs focussing on the degradation of ILs under US. In the past studies, the degradation after hours was reported, together with the various formed degradation products. The evolution of SL spectra was also reported but with a time-resolution of 1 minute. The novelty of the present study is the demonstration of the feasibility of measurement of SL spectra in the very first moments of sonication, which allowed to bring to light that degradation starts to occur from the onset of cavitation. We agree that transient effects can arise in this initial stage, and that’s what we report here as “fluctuations”. However, it is also noticeable that present results agree with previous ones.

  1. “In line 85, the author states that the temperature of the IL is controlled at 25°C. Then how does the temperature increase as shown in Fig. 2? Is the temperature controlling unit stopped during sonication? This is just an example of the many similar issues appearing in the manuscript. Please clarify the test condition for each dataset you analysed.”

Some more details have been added in the experimental section to clarify this point (see question 2). The cryostat temperature is set at 25°C and the cryostat is kept on during sonication. So, initial temperature of the IL is 25°C. As soon as cavitation takes place, the IL is heated by it, because a large part of the acoustic energy is transformed into heat (as is well known in the US community).

  1. “line 122, if temperature control is not used in the test, why does the temperature pleateau at 45°C?”

Temperature plateauing is due to the cooling from the cryostat and the heating from cavitation. A sentence has been added in section 2: “Thus, the IL initial temperature is 25°C and then increases during the sonolysis due to cavitation.”

  1. “line 256-259. The author analyses how high viscosity may lead to degradation of IL by forming large SL bubbles which are stable under high acoustic pressures. However, in Fig. 5, the viscosity of dry IL at 14 s barely changes from the initial value. If degradation appears at this instant, I would argue similar degradation would occur at 0 s according to the same reasoning. In addition, I doubt at high viscosity, the bubble can be large enough to be able to make translational move.”

We believe there is some misunderstanding here. We agree with the reviewer that similar degradation occurs at 0 s indeed, and this is stated in the manuscript at several occurrences for instance in the abstract: “IL degradation is observed as soon as SL emission appears”.

Concerning the translational move, it is something observed by other authors at high viscosity. However, it is only one phenomenon that could explain the jetting of bubbles and subsequent degradation of IL inside cavitation bubbles, in particular due to stronger secondary Bjerkness forces and/or pecular bubble structures formed. This is discussed in 3.3.

  1. “It seems from Fig.6 that the molecular emission around 390 nm disappears at 99 s (the top spectrum line). Please explain this phenomenon.”

The reviewer is right, this emission appears not to be visible any more after a certain time. This is due to light absorption, as indicated in 3.2: “progressive colouring of the IL occurs, leading to strong absorption of the SL light in the UV and visible domains”.

We would like to thank again the referee for his/her comments and hope that the revised version is now acceptable for publication in Molecules.

We would like to thank again the referee for his/her comments and hope that the revised version is now acceptable for publication in Molecules.

Round 2

Reviewer 2 Report

The authors largely addressed my concerns except the one regarding data repeatability. What I would like to see is, taking Fig.3 as an example, multiple tests showing that new S2 emission bands did appear around 51 s. Nevertheless, I understand that the reported conclusion is more about qualitative, rather than quantitative. So, I considered this issue is not significant.